# The Impact of the Built Environment and Social Environment on Physical Activity: A Scoping Review

**DOI:** 10.3390/ijerph20126189

**Published:** 2023-06-20

**Authors:** Yiyu Wang, Bert Steenbergen, Erwin van der Krabben, Henk-Jan Kooij, Kevin Raaphorst, Remco Hoekman

**Affiliations:** 1Behavioral Science Institute, Radboud University, 6525 XZ Nijmegen, The Netherlands; bert.steenbergen@ru.nl; 2Department of Geography, Planning, and Environment, Institute for Management Research, Radboud University, 6525 XZ Nijmegen, The Netherlands; erwin.vanderkrabben@ru.nl (E.v.d.K.); henk-jan.kooij@ru.nl (H.-J.K.); kevin.raaphorst@ru.nl (K.R.); 3Mulier Institute, 3584 AA Utrecht, The Netherlands; r.hoekman@mulierinstituut.nl

**Keywords:** physical activity, built environment, social environment

## Abstract

This scoping review aims to provide an overview of the current state of physical activity research, focusing on the interplay between built and social environments and their respective influences on physical activity. We comprehensively searched electronic databases to identify relevant studies published between 2000 and 2022. A total of 35 articles have been reviewed based on the research question. The review found that built and social environments influence physical activity, and consideration of people’s perceptions of their surroundings can provide further insight. The literature was summarized, and recommendations were made for future research. Findings suggest that interventions targeting built and social environments can promote physical activity effectively. However, limitations in the literature exist, including a need for more standardization in research methods and consistency in measurement tools.

## 1. Introduction

Moderate-to-vigorous physical activity (PA) is critical to reducing chronic diseases such as obesity and enhancing the population’s health. Although physical activity has raised more attention in the past years, public health statistics still show that, globally, 28% of adults aged 18 and over were not active enough to reach the WHO’s recommendation in 2016 (men, 23%; women, 32%) [1]. This growing development of insufficient physical activity has, in turn, exacerbated obesity, cardiovascular disease, diabetes, and other chronic diseases. These chronic diseases account for approximately 70% of deaths globally [1]. In the context of inactivity levels and the recent COVID-19 pandemic, research on physical activity and how to promote it is of great importance.

One of the many factors for improving physical activity is the built environment. The built environment is essential to the physical environment, including urban and architectural design, land use, transportation, and corresponding infrastructure support facilities [2]. For example, studies show that mixed-use residential, commercial, office, entertainment, and other land use can significantly increase walking [3]. In addition, the street network pattern can influence the choice of travel routes and modes of transportation. For example, high connectivity encourages active transportation by reducing travel distances while providing multiple travel route choices [4,5]. Furthermore, good accessibility and connectivity of destinations are beneficial in increasing opportunities for people to walk/bike commute or travel daily [6]. Based on the existing research on the built environment and physical activity and the evidence from this research, the built environment has become a critical intervention and policy tool promoting physical activity.

Relevant policies involving urban planning and public transportation are closely related to the level of participation of the public in physical activities. In urban planning, for the management of land-use types, relevant laws and policies can be formulated to require that a certain amount and proportion of land be used for the construction of physical activity facilities to ensure that sufficient venues are provided for people’s physical activities [7]. For transport intervention, paving sidewalks and bicycle lanes in cities will reduce urban traffic congestion and environmental pollution caused by car exhaust and help more people choose physical activities such as walking and cycling [8]. However, it is challenging to predict physical activity patterns in the built environment simply by the friendliness of the built environment. Individual physical activity requires not only the support of the built environment but also the support of the social environment.

The social environment includes individual factors (including age, gender, fitness, or biology such as genetic predispositions and neurological development, etc.), social networks (including family, peers, school, community, and work), and a wide range of background conditions (such as culture and economy) [9]. Several studies have examined the relationship between critical demographic variables such as gender, income level, education level, and physical activity. Some research states that men are more involved in physical activities than women, particularly in high-income countries [10,11,12,13]. In addition, cultural norms around gender roles and expectations can also influence women’s engagement in physical activity, particularly in more conservative societies [14]. Others have reported that in some African countries, such as Mozambique, Tanzania, and Uganda, women tend to engage in more physical activity than men [15]. Studies have found that in Asian countries, such as China, women tend to engage in more physical activity than men [16]. In addition, individuals with higher income levels [17,18] and higher education levels are more likely to participate in leisure-time physical activities [19,20].

Research on the factors that affect physical activity individuals has received increased attention in urban planning, sociology, and behavior research. With the introduction of the social ecological model of health behavior [21], research on the influencing factors of behavior is no longer limited to the individual level of age, gender, knowledge, and skills. Instead, it has become multifactorial, also including aspects of work, the built environment, and leisure time. Its core feature is that physical activity results from multiple factors [22]. Bronfenbrenner’s social ecological theoretical model provides a comprehensive and interdisciplinary analysis framework, broadens the research ideas of physical activity-related factor analysis, and has been widely used in empirical research. In addition, it exemplifies the urban built environment’s influence on physical activity [23,24].

However, compared to the abundance of research that uses the social ecological model to study the influence of the built environment on physical activity, research on the intricate relation between built and social environments and their combined influence on physical activity is relatively scarce. As a notable exception, a study in Hong Kong takes housing type as an indicator and found that this aspect of the built environment is a significant determinant of social environments, social contacts, and activity-travel behavior [25]. Similar research in the US also found that housing type limits options for a healthy lifestyle and social cohesion, which in turn influences physical activity [26]. From the perspective of research content, there is an excellent potential for research on the relationship between built and social environmental factors that affect physical activity and types of physical activity. This necessitates the systematic exploration of physical activity research by integrating the urban planning discipline with sociology and behavioral approaches.

We focus on filling this knowledge gap in the present scoping review. Specifically, we examine the state of the art in physical activity research, which includes the interaction between the built environment and the social environment and the advantages of combining built and social environments and the impact of this combination on physical activity research. This review provides a solid base and lays bare areas of future research that combine the abovementioned disciplines. Finally, we discuss the results in light of the potential advantages of interdisciplinary approaches for promoting physical activity.

## 2. Methods

The research adopts the [27] scoping review research framework, divided into five steps: (1) Clarify the research question; (2) Determine the relevant research by searching PubMed, EMBASE, PsycINFO, and Web of Science databases; (3) Screen the target literature; (4) Data extraction, i.e., according to the research purpose, develop an information extraction table to incorporate critical research information, including basic information such as published author, year, and country, as well as research process information such as research purpose, research design, data extraction method, and sampling method. Relevant information also includes research results such as outcome measurement, methods, and leading research results; and (5) summarize and present the results. This research uses thematic analysis to classify the themes mentioned above and presents findings as tables and summary paragraphs. The following graphic describes the procedure of data retrieval, including the number of articles retrieved in each stage of the process. See Figure 1. The data extraction process.

### 2.1. The Research Questions

Based on the questions addressed in the introduction, this scoping review defines the focused research questions as follows:(1)To what extent has current research included both the built environment and the social environment as factors that impact physical activity?(2)What can we learn from this research concerning the advantages of the combination of built and social environments and the impact of this combination on physical activity?

### 2.2. Searching for Relevant Research

Literature retrieval and reporting methods follow the requirements of scoping reviews. An information specialist of the library was consulted for the evaluation setup, which included choosing the right databases, suitable keywords, and mesh terms. Literature data were collected through Web of Science, PubMed, EMBASE, PsycInfo database, and Google Scholar. The three primary keywords were the built environment, social environment, and physical activity.

#### 2.2.1. Physical Activity

Physical activity mainly refers to any physical actions that require energy consumption produced by skeletal muscles, including work, housework, transportation, leisure, and other activities [28]. The measurement of physical activity level mainly adopts self-reporting methods and, to a lesser extent, physical activity trackers. Physical activity outcomes include traffic-related walking, recreational walking, cycling, moderate-to-vigorous-intensity physical activity (MVPA), and total physical activity.

#### 2.2.2. Built Environment

The built environment is an essential part of the physical environment, including urban and architectural design, land use, transportation, and corresponding infrastructure support facilities. The built environment is defined as the objective and subjective characteristics of the physical context in which people spend their time (e.g., home, neighborhood), including aspects of urban design, (e.g., presence of sidewalks), traffic density and speed, distance to and design of venues for physical activity (PA) (e.g., parks), and crime and safety [29].

#### 2.2.3. Social Environment

According to the social ecological model, human health is determined by individual factors (including age, gender, fitness, or biology such as genetic predispositions and neurological development, etc.), social networks (family, peers, school, community, work), environment (built environment, social environment, policy), and a wide range of background conditions (such as culture and economy) [9]. The social environment refers to the relationships, culture, and society that individuals interact with, including social influencers such as friends and family [30].

#### 2.2.4. Search Strategy

An information specialist performed the search on 30 March 2020, and the search formula comprised the main search terms (topic terms) connected by logical words “AND” and “OR”. For example, on PubMed, the keywords and mesh terms that were chosen are: ((Exercise [MesH] OR Bicycling [MesH] OR Sedentary behavior [MesH] OR physical activit* [tiab] OR exercise* [tiab] OR walking [tiab] OR bicycling [tiab] OR cycling [tiab] OR sedentary behavior [tiab] OR sedentary behaviour [tiab]) AND (Environment Design [MesH] OR environment design* [tiab] OR Urban planning [tiab] OR urban environment* [tiab] OR built environment* [tiab] OR physical environment* [tiab] OR healthy place* [tiab] OR universal design* [tiab] OR human centered design* [tiab] OR design for all [tiab])) AND (Social environment [MesH] OR social environment* [tiab] OR social context* [tiab] OR social ecology* [tiab] OR community network* [tiab] OR community health network* [tiab] OR social support [tiab]). See the search strategy for EMBASE, PsycInfo, and Web of science in Note 1.

Note 1: Search strategy

(1)PubMed

#1 Physical activity/Walking/Cycling

Exercise [MesH] OR Bicycling [MesH] OR Sports [MesH] OR Sedentary behavior [MesH] OR physical activit* [tiab] OR exercise* [tiab] OR walking [tiab] OR bicycling [tiab] OR cycling [tiab] OR sedentary behavior [tiab] OR sedentary behaviour [tiab] OR sport* [tiab]

#2 Built Environment—urban planning intervention to promote active living at street level, community/neighborhood level

Environment Design [MesH] OR environment design* [tiab] OR Urban planning [tiab] OR urban environment* [tiab] OR built environment* [tiab] OR physical environment* [tiab] OR safety [tiab] OR healthy place* [tiab] OR universal design* [tiab] OR human centered design* [tiab] OR design for all [tiab] OR land use [tiab] OR land usage [tiab]

#3 Social Environment

Social environment [MesH] OR Interpersonal relations [MesH] OR social environment* [tiab] OR social context* [tiab] OR social ecology* [tiab] OR community network* [tiab] OR community health network* [tiab] OR social support [tiab] OR safety [tiab] OR social relation* [tiab] OR interpersonal relation* [tiab]

(2)Embase

#1 Physical activity/Walking/Cycling

Exp exercise/OR Exp physical activity/OR exp sports/OR exp sedentary lifestyle/OR (physical activit* OR exercise* OR walking OR bicycling OR cycling OR sedentary behavior OR sedentary behavior OR sport*).ti,ab,kw.

#2 Built Environment—urban planning intervention to promote active living at street level, community/neighborhood level

Exp environmental planning/OR (environment design* OR Urban planning OR urban environment* OR built environment* OR physical environment* OR healthy place* OR universal design* OR human centered design* OR design for all OR land usage).ti,ab,kw.

#3 Social Environment

Exp social environment/OR (social environment* OR social context* OR social ecology* OR community network* OR community health network* OR social support safety OR social relation* OR interpersonal relation*).ti,ab,kw.

(3)PsycInfo

#1 Physical activity/Walking/Cycling

Exp exercise/OR Exp physical activity/OR exp sports/OR exp sedentary lifestyle/OR (physical activit* OR exercise* OR walking OR bicycling OR cycling OR sedentary behavior OR sedentary behavior OR sport*).ti,ab,id.

#2 Built Environment—urban planning intervention to promote active living at street level, community/neighborhood level

Exp environmental planning/OR (environment design* OR Urban planning OR urban environment* OR built environment* OR physical environment* OR healthy place* OR universal design* OR human centered design* OR design for all OR land usage).ti,ab,id.

#3 Social Environment

Exp social environment/OR (social environment* OR social context* OR social ecology* OR community network* OR community health network* OR social support safety OR social relation* OR interpersonal relation*).ti,ab,id.

(4)Web of Science

#1 Physical activity/Walking/Cycling

TOPIC: (“physical activit*” OR exercise* OR walking OR bicycling OR cycling OR “sedentary behavior” OR “sedentary behaviour” OR sport*)

#2 Built Environment—urban planning intervention to promote active living at street level, community/neighborhood level

TOPIC: (“environment design*” OR “Urban planning” OR “urban environment*” OR “built environment*” OR “physical environment*” OR “healthy place*” OR “universal design*” OR “human centered design*” OR “design for all” OR “land use” OR “land usage”)

#3 Social Environment

TOPIC: (“social environment*” OR “social context*” OR “social ecology*” OR “community network*” OR “community health network*” OR “social support” OR “safety” OR “social relation” OR “interpersonal relation”)

### 2.3. Screening the Target Literature

Literature screening was performed in three consecutive steps: (1) remove duplications, (2) screen the article’s abstract, and (3) set criteria for eligible articles.

The criteria were set based on the research questions. To answer the research questions, the selected article was required to contain three key concepts, which are physical activity, the built environment, and the social environment.

After discussion with the team members, the criteria of the eligible articles were set as follows: (1) includes objective built environment measurements; (2) includes social environment measurements; (3) includes physical activity measurements such as traffic-related walking, recreational walking, cycling, moderate-to-vigorous-intensity physical activity (MVPA), and total physical activity; (4) includes a human population.

The criteria for non-eligible articles were set as follows: (1) only includes an analysis of the built environment’s impact on physical activity; (2) only includes an analysis of the impact of the social environment on physical activity; (3) only involves disease-related physical activity; (4) does not perform in the real world; (5) does not include objective environment measurement; (6) does not include physical activity.

### 2.4. Data Extraction

According to the research purpose, the research team created a data extraction table that includes main characteristics incorporating the selected articles’ critical information. Data extraction includes the following: (1) research aim, (2) physical activity measurements, (3) type of physical activity, (4) built environment measurements, (5) social environment measurements, (6) data sample, (7) population demographic, (8) location, (9) analytical method, and (10) findings.

### 2.5. Summarizing and Presenting the Results

To summarize the results, this research used thematic analysis to classify the abovementioned themes and report them in tables and summary paragraphs. First, we created a table that provides an overview of the characteristics of the selected articles (Table A1). Second, we focused on the research combining the built and social environments’ impacts on physical activities. Finally, we discussed the results and provided suggestions for future studies.

## 3. Results

### 3.1. Overview of the Studies—Table A1 Provides an Overview of the Characteristics of the Selected Studies (See Appendix A)

#### 3.1.1. Demographic Characteristics

The 35 selected studies showed a great variety of research populations. Of the 35 included studies, 22 studies had adults as participants (n = 63%). Thirteen studies (n = 35.37%) had adolescents as participants. Eight studies focused on females of different ethnic minority groups (n = 23%). The final three studies were conducted on the elderly population (n = 8%).

#### 3.1.2. Geographic Characteristics

The selected studies covered geographic diversification. Thirteen studies were conducted in the US (n = 37%). A further 10 studies were conducted in European countries (n = 28.5%) and 6 studies were performed in Australia (n = 17%). Finally, three studies from Asian countries (n = 8%) and two from Brazil (n = 5%) were found.

#### 3.1.3. Social Environment Characteristics

Studies examined various aspects of the social environment. At the individual level, demographic characteristics were evaluated, and SES (socioeconomic status) is one of the most evaluated characteristics. Of the 35 studies, 30 studies evaluated the SES of the participants (n = 86%), and 5 of the 30 studies had a research focus on low-SES communities. At the community level, the most evaluated factor is community/neighborhood safety (14/35). In addition, general social support (10/35) is given much attention. Finally, social cohesion (8/35) has also been highlighted.

#### 3.1.4. Built Environment Characteristics

Concerning aspects of the built environment, of the total 35 studies, the majority of articles (n = 30.85%) focused on community. Regarding more specific urban settings, five studies (n = 15%) looked into the school environment. Researchers have evaluated various built environment factors in the papers that we have reviewed. Accessibility (20/35) and connectivity (18/35) have received the most attention, while other studies have also incorporated environment perception into evaluation of the built environment (15/35).

#### 3.1.5. Physical Activity Characteristics

Physical activity was divided into commuting, leisure, and total physical activity. For commuting-related physical activity, four studies (n = 35.11%) measured the daily commutes (walking or cycling) to transportation hubs and final destinations. Twenty-six studies (n = 75%) focused on leisure-time physical activity estimates, such as minutes of moderate-to-vigorous physical activity per day or week. The remaining five studies (n = 14%) reported the total physical activity in minutes.

#### 3.1.6. Measurement Characteristics

All 35 selected studies measured the perception of the social environment through surveys/questionnaires/interviews. These methods were different for the built environment. Of the 35 studies, 11 (n = 31%) used GIS (geographic information systems) to provide an objective measurement of aspects of the built environment. Fourteen studies (n = 40%) used surveys/questionnaires/interviews regarding perceptions of the built environment to represent the subjective measure of the physical environment in which participants lived. Children’s perception of the neighborhood was measured with the help of their parents. One study used mapping exercises to determine the perception of a neighborhood. Here, participants were asked to use colored pens to identify key routes and destinations related to their walking activities [31]. Among the 35 studies, the physical activity data of 29 studies were obtained through self-reported questionnaires. Six studies applied physical activity tracking devices to participants. Four of the latter six studies combined questionnaires and direct activity monitoring.

### 3.2. Social Environmental Impact on Physical Activity

#### 3.2.1. Interpersonal Level

At the interpersonal level, participation and persistence are essential for physical activity, and social support is an important variable that promotes people’s participation in physical activity. Individual social support generally comes from family and peer support. Three of thirty-five studies showed that support for physical activity from closely related persons positively impacts an individual’s physical activity behavior at all ages [32,33,34]. In three studies with adolescents, it was shown that family, friends, and peers are the sources of social support affecting participation in physical activity [31,35,36]. Notably, in two studies with children, participation in physical activity, occupation, and education level of parents and other family members were used as independent variables to examine children’s physical activity [36,37]. Social interactions were also considered vital in two studies, one with children and adolescents [33] and another with adults [36]. Children and adolescents who hang out and play with their friends are more prone to physical activity during the weekdays [33,36]. One study also suggested that physical activity role models promote people’s participation in physical activity [38].

#### 3.2.2. Community Level

At the community level, the safety of the community has been studied. Five studies used local crime rates to determine safety [38] and four studies applied perceived neighborhood safety surveys [33,39,40,41]. One study found that people are more engaged in physical activity if the community’s safety can be guaranteed. Conversely, if people’s perceived community environment is unsafe, it will lead to decreased physical activity, a decline in health and fitness levels, and social isolation [42]. In addition, social cohesion was mentioned in eight studies. This aspect is used to examine the sense of belonging [43], trust [44], value [43], and healthy beliefs [45]. Moreover, social norms can also change people’s physical activity and health behaviors. Social norms reflect standards of behavior and generally accepted values in society. For example, one study’s results showed that if it is common in a neighborhood for people to walk, then people in the neighborhood are more likely to walk to a travel destination [43]. In studies with female participants, the results also indicated that seeing people exercise in the neighborhood was positively related to higher levels of physical activity [43,46,47,48,49].

#### 3.2.3. Other Aspects

In addition to the aspects mentioned above, three studies from Australia examined the ownership of sports membership cards [22,32,45], and one study looked into dog ownership [50]. The study provided evidence that the ownership of dogs positively impacted an individual’s physical activity level. Another two studies showed a lack of positive impact of sports membership on adults’ physical activity [22,32]. Finally, one study from the US showed that participation in a recreational program with friends was associated with an increased likelihood of the elderly’s physical activity [44].

### 3.3. Built Environmental Impact on Physical Activity

Researchers examined a variety of built environmental features in the selected studies.

#### 3.3.1. Accessibility

Three studies have pointed out that objective accessibility between residences and buses or subway stations is significantly correlated to an increase in the daily walking/cycling commuting of residents [36,51,52]. In addition, three studies on adolescents [35,39,53] found that the closer the a destination (including parks, workplaces, and commercial facilities), the greater the opportunity for transformational walking among adolescents. Similar to objective accessibility, subjectively perceived accessibility is positively correlated with residents’ physical activity level. Six studies showed that the better the perceived distance of residents to public transport stops or destinations, the more likely they are to increase walking or cycling opportunities [42,43,46,47,48,49,54] and recreational physical activity frequency [39].

#### 3.3.2. Connectivity

Five studies primarily examined the relationship between road network connectivity and walking or cycling behavior by measuring neighborhood street length, area, intersection density, street density, and other indicators [22,32,36,50]. Nine studies showed that the shorter the distance to a destination, the more likely it is that commuting via walking or cycling to that destination is increased in adults or children [31,42,43,46,47,48,49,55]. One study also showed this for recreational walking and commuting via walking in an elderly population [56]. In addition, three studies suggested that the better adults perceive neighborhood street connectivity, the higher the probability of using walking or cycling as a mode of transportation [57,58], and the more reassured adolescents are about taking active modes of transportation to and from school [44,51].

#### 3.3.3. Built Environmental Quality

Objective built environmental quality: These studies focus on the impact of the comprehensive functional quality of the pedestrian environment and the functional quality of a single public open space. Five studies found that the higher the quality of the neighborhood walking environment, the easier it is to increase the probability of active travel by children [51], to increase the likelihood of active travel by adults [59], to increase levels of adult traffic walking or cycling [57], to increase recreational physical activity [60], and to increase moderate-to-high-intensity physical activity [41]. In addition, seven studies pointed out that the attractiveness, safety, convenience, maintenance, diversity, and high-quality characteristics of public open spaces, especially parks, significantly affect physical activity and lead to more leisure-time physical activity [22,32,35,50,61,62,63].

Subjective built environment quality: Based on exploring the impact of objective built environmental quality and the interaction between individual daily physical activities and built environmental elements, five studies incorporate individual subjective feelings into their analyses, focusing on perceived effects of the walkability, aesthetics, safety, and maintenance of the neighborhood built environment [45,52,58,62,64]. Three studies found a correlation between perception of the built environment and physical activity [33,34,44]. Two of the three studies showed that residents have stronger perceptions of walkability and aesthetics in their neighborhood or public open spaces and show higher levels of recreational activity and are more likely to achieve the recommended amount of physical activity [34,44]. In addition, 10 studies showed that the perception of safety factors related to crime and traffic might affect residents’ willingness to engage in outdoor activities, thereby affecting their traffic and leisure activity levels [31,42,43,45,46,47,48,49,55].

## 4. Discussion

This scoping review aims to analyze the state of the art in physical activity research, which includes the interaction between the built environment and the social environment and the impact of the combination of built and social environments on physical activity.

First, we discuss the impact of social and built environments on physical activity. Second, we discuss the relationship and interaction between social and built environments in physical activity research. Finally, we discuss the results in light of the potential advantages of interdisciplinary approaches for promoting physical activity.

### 4.1. Influence of Individual, Social, and Built Environmental Characteristics on Physical Activity

In this section, we further discuss the main result in detail. First, we have already noted that a large number of studies have focused on the adolescent population. In previous behavioral research, adolescence has been regarded as a critical period for behavioral change [65]. Furthermore, studies have shown that exercise habits developed during childhood and adolescence are highly likely to carry over into adulthood [66,67]. Therefore, promoting the habit of participating in physical activity in adolescence is particularly important in one’s life. This may explain researchers’ primary focus on physical activity research in adolescents.

Another research trend that we found is physical activity studies of females. The possible reason behind this is that research has documented that females are less physically active than males [68,69,70,71,72] and less likely to participate in physical activity outdoors due to safety concerns [73]. These barriers can potentially lead to inactivity and poor health outcomes. Therefore, females have been identified as a high-priority group for physical activity interventions [74].

In our selected articles, no studies focused on male groups alone. As some studies on men’s health behavior point out, a possible reason is that males have overall been underrepresented in physical activity intervention studies [75].

#### 4.1.1. Social Environment

In our review, the social environment was shown to strongly influence the physical activity of all age groups, in particular, aspects of social interaction, social support, and social cohesion.

Our review showed that in the adolescent studies, the social environment, especially aspects of parental values, parental constraints, and interaction between parents and adolescents, influences their physical activity pattern [36,37,42,52]. For example, a higher degree of family participation in physical activity is positively correlated with adolescent physical activity [36]. These findings align with previous research that identified family structure, parental values, educational level, and work situation indirectly affecting children’s physical activity behavior [76,77]. In line with this, previous research showed that having social interaction with peers can be a reason that adolescents are physically active [78]. Our review results confirm this by showing similar evidence that being able to hang out with friends and peers in the neighborhood can motivate adolescents’ physical activity behavior [33].

Another crucial social environmental factor for adolescents’ physical activity is social support. Social support has been considered a buffer against physical activity decline during the transitional period [78]. Our results show that verbal or active support for adolescents’ physical activity from family [37], friends [36], and peers [31,33,35] directly affects adolescents’ physical activity behavior. For example, adolescents who received more support for physical activity from friends participated in physical activity more often [36]. Four of the thirteen selected studies on adolescents indicate that friends and peers are the most influential source of social support for adolescent participation in physical activity [26,31,33,36]. This might be explained by the fact that adolescents spend most of their time in school, and adolescents in this age group gradually receive less parental supervision and more peer influence [73,79,80].

Social support is also an essential influencing factor for other demographic groups. For example, our results indicate that women who receive support from their families for physical activity are more likely to reach the recommended physical activity level compared to those who do not [46,47,48,49,55,64]. There is also evidence that social support may be more influential for women, especially the support they receive from their family [81,82,83]. Similar evidence was also found in a study with older people, in which the social environment was considered a more critical influencing factor of physical activity than the built environment [84]. For example, older people with neighbor/family member relations can take a walk, providing a better chance of achieving the recommended level of physical activity [40,45]. Previous studies are in line with these findings that social support from family, friends, and community can motivate positive physical activity behaviors in various demographic groups [85,86].

Social cohesion is generally defined as building shared values and making people feel engaged in common causes, facing challenges, and being members of the same community [87]. Our review showed that social cohesion impacts the physical activity of all demographic groups. For example, at the community level, higher social cohesion is associated with lower crime rates, and lower community crime rates tend to be associated with greater participation in physical activity [38,56,57,64].

#### 4.1.2. Built Environment

Our review shows that researchers have evaluated various built environmental factors. Accessibility and connectivity have received most of the attention. This might be because these two factors are directly measurable via geo-data, are easy to quantify and standardize, and because research results are easy to translate into policy [88]. However, GIS-driven geo-data also have limitations as objective geo-data cannot fully reflect the quality of the built environment [89].

In our review, it is shown that a number of researchers have started to investigate subjective perceptions of accessibility, connectivity, and built environmental quality. This might be because it was clear that the built environment’s impact on an individual’s physical activity behavior cannot be fully explained by objectively measured indicators [89,90]. For example, our findings show that perceived access was more important than objective measures of park access (tract-level park count, distance to nearest park, percent of tract covered by parks) for children and adolescents [36]. In addition, different resident groups may have different perceptions of the same environment and facilities [91]. For example, people with higher green space requirements and expectations usually think that greenery conditions in their neighborhood are bad, and they are concerned about outdoor activity experiences. In comparison, people with lower requirements and expectations may be more optimistic and believe that the greenery is of good quality [39]. Environments with the same objective measurement results can, thus, have different health effects on different groups.

### 4.2. Interaction and Relationship between Social and Built Environments

#### 4.2.1. Interaction

Although all selected studies included both social and built environments, only one study discussed the interaction between social and built environments. This study aimed to investigate whether social and built environmental interactions are associated with physical activity in underprivileged communities in the UK [62]. The research result suggests that the social environment moderates the built environment’s impact on physical activity. For example, the findings suggest that community cohesion and safety moderate the impact of physical barriers (e.g., the destruction of public spaces and buildings) on residents’ walking behavior, i.e., when residents perceive their community as having a higher level of integration. The damage to public spaces and buildings can affect residents’ physical activity. In addition, social interaction moderates the impact of aesthetics of the built environment on physical activity; for example, when residents have a higher level of social interaction, the aesthetics of the built environment affect individual physical activity behaviors. These findings suggest that aspects of the social environment may be more critical than physical aspects in encouraging individuals to be active in deprived environments [62].

As a concluding statement with regard to this topic, we can argue that community-level and individual-level socio-environmental factors influence individual physical activity outcomes and moderate the link between built environmental factors and physical activity. However, the studies included in this article only discuss the synergistic effects of social and built environments in a deprived neighborhood on residents’ physical activity. We also require more research and evidence across different community types to further define the interaction between society and the built environment. For example, in affluent and family-based communities, we need to examine whether the physical activity of residents in these communities is also affected by the synergistic effect of the social environment and the built environment.

#### 4.2.2. Relationship

Regarding the relationship between social and built environments in physical activity research, we have found a collaborative relationship between these two environments in our review. We found that social and built environments both impact physical activity. Together, they form the neighborhood individuals are situated in, and after processing the information, the individual creates a neighborhood perception, which can also influence an individual’s PA pattern [31,58,62,64] (Figure 2).

Researchers have proposed that environmental perception is the psychological environment formed by the perceiver after receiving and processing information in the physical environment, which could guide external behavior [92]. For example, for residents who were satisfied with objective environmental quality and neighborhood safety, their perceptions of walkability and safety in their neighborhood or public open spaces were higher, and they were more likely to achieve the recommended amount of physical activity [58]. Furthermore, our selected studies of women showed a weak correlation between physical activity and the built environment compared to the perceived environment. In particular, the perception of good street lighting at night was a significant correlate of physical activity [43,47]. However, in contrast with this, some studies show that the aesthetic perception of the neighborhood environment is negatively related to the level of physical activity, especially with regard to walking [93,94]. Possible reasons for the inconsistent conclusions are that attractive neighborhoods with low mixed land use, functional inconvenience, and potentially poor street connectivity do not facilitate transportation-related physical activity for residents [95].

Generally, neighborhood perception as the collaborative result of social and built environments has become a new hotspot in physical activity research. Our findings show that neighborhood perception can help us capture specific groups’ concerns about specific social and built environmental contexts. For example, women are sensitive to the presence of street lights at night [43,46,47], parents are concerned about traffic safety [44], and older people are concerned about social interaction with community partners [40,45]. Through neighborhood perception, we can explore the influencing factors of physical activity from participants’ perspectives. Researchers have suggested that more research should focus on participants’ viewpoints of the environment [96]. However, there is still much to learn in this area, and further research is needed to deepen our understanding of the relationship between environmental perceptions and physical activity.

### 4.3. Potential Advantages of Interdisciplinary Approaches for Promoting Physical Activity

All 35 selected articles use social and built environments to understand an individual’s physical activity level. This shows that researchers are increasingly paying attention to the importance of studying and promoting PA from an interdisciplinary perspective. Our review suggests that social context can further explain physical activity behaviors in adolescents, women, older adults, and those from disadvantaged communities [31,33,35,42,45,46,47]. These findings are in line with the suggestion of previous studies that applying individual social environments to construct environmental analysis can better distinguish the influencing factors of physical activity among residents of different socioeconomic status, socio-demographic characteristics, and community types [97]. In addition to improving research accuracy, our review suggests that including measurement of the social environment in physical activity research may help tailor interventions to promote future physical activity. For example, the intervention design for women’s physical activity can start from their social environment; our result showed that participation in community activities is positively correlated with women’s physical activity [43,47,48,49,50,54]. Another example is deprived neighborhoods; our results demonstrate that social cohesion and interaction influence the built environment’s impact on an individual’s physical activity. Therefore, the design of physical activity interventions for poor communities should promote community participation in plan-making processes and increase the interaction between community residents [62].

### 4.4. Suggestions for Future Study

We have found a critical knowledge gap regarding the interaction of social and built environments. Our review found that social and built environments substantially impact an individual’s physical activity. In addition, researchers suggest that social and built environmental characteristics’ interactive effect may also impact an individual’s physical activity [97]. According to the social ecological model, human health is determined by individual factors, social networks, the built environment, and overall background conditions [98]. Furthermore, the social ecological approach to physical activity argues that individual characteristics, social environment, physical environment, and policies are all critical determinants of physical activity, which are interconnected and embedded in complex systems [99]. Therefore, understanding the interaction between these two factors may help us understand the synergy of social and built environments and utilize these two measures more effectively [100]. Therefore, we need more studies that analyze the interaction between social and built environments and the relevant impacts on physical activity.

The review result indicates that researchers fully recognize the importance of environmental perceptions in physical activity, and more literature has begun to take participants’ viewpoints on social and built environments as essential variables that impact physical activity [101,102,103]. Many recent studies applied neighborhood perception as the indicator, but in reality, perception elements are challenging to capture and require high accuracy [93]. To address this issue, researchers must continue to explore innovative methodologies and measurement tools that can provide a more accurate understanding of the impact of environmental perceptions on physical activity.

## 5. Conclusions

After conducting a thorough review of recent physical activity research, this scoping review has found evidence supporting the notion that both built and social environments can influence physical activity levels. The literature reviewed suggests that having access to safe and convenient infrastructure, as well as social support from friends and family, can increase the likelihood of engaging in regular physical activity.

Moving forward, it is essential for researchers to continue exploring participants’ perspectives on built and social environments in order to develop effective strategies to promote physical activity and improve public health outcomes. By combining insights from both built and social environments, policymakers and public health officials can develop more comprehensive and effective interventions to promote physical activity at the population level.

## Figures and Tables

**Figure 1 ijerph-20-06189-f001:**
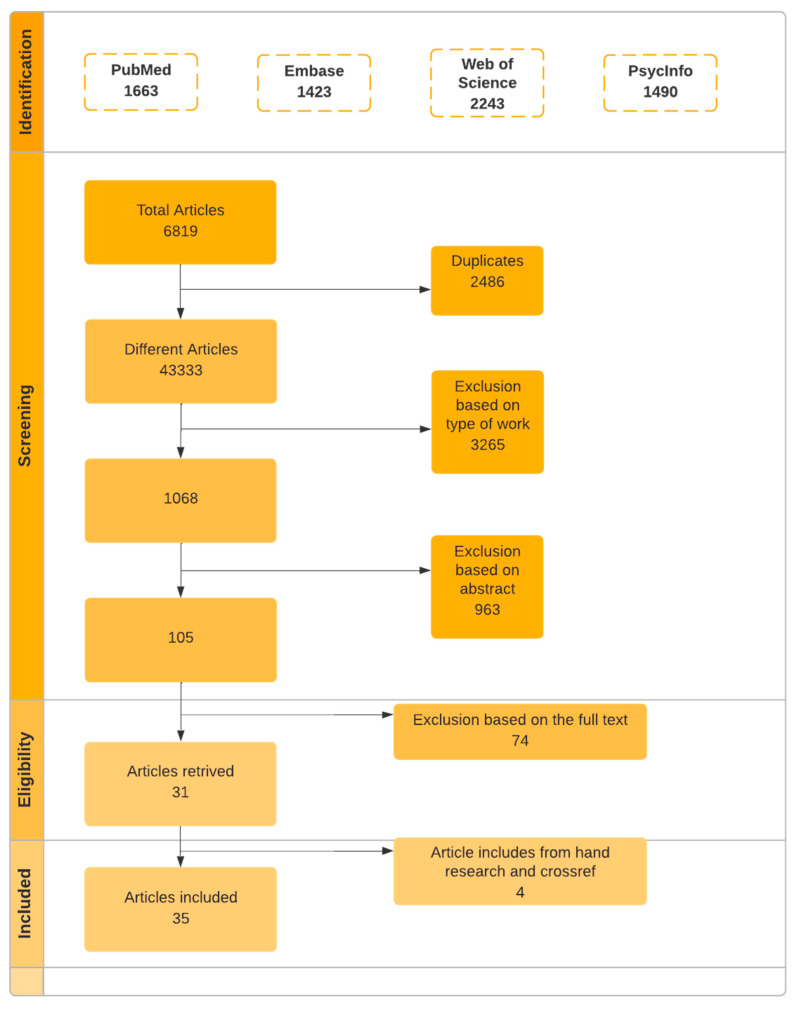
The data extraction process.

**Figure 2 ijerph-20-06189-f002:**
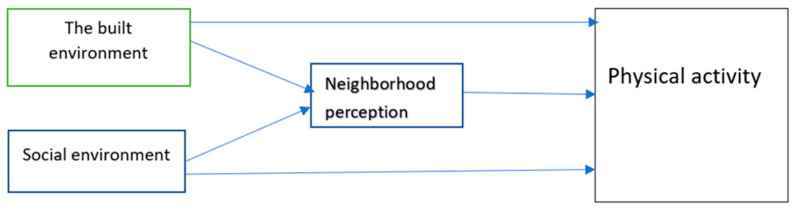
The relationship between built and social environments and neighborhood perception and their impacts on physical activity.

## Data Availability

Not available.

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
