# Peer review of "The Impact of the Built Environment and Social Environment on Physical Activity: A Scoping Review"

_ijerph, 2023, doi:10.3390/ijerph20126189_

Round 1

Reviewer 1 Report

Dear Authors,

The paper is very interesting and relevant.

I would like to make a few comments.

Major Issue

In the introduction and in point 2.2.1 it talks about studies indicating that higher income levels and higher levels of education are more likely to be physically active. They are probably referring to physical activity outside of work. Evidently, these people with high income and high educational levels tend to have more sedentary jobs.

I think it should be better explained what the authors mean by physical activity because there are jobs where physical activity is large.

Regarding the fact that women tend to be less physically active I think it should be discussed more because this fact will depend on social class, economic, country, etc.

Minor issue

In the sentence "The following graphic describes the procedure of data retrieving, including the number 110 of articles retrieved in each stage of the process." there should be a "reference" to the graphic.

Reviewer 2 Report

Dear Authors,

            I read your manuscript with interest. The study under review is up-to-date and it is obvious that you have worked very hard to conceive and write it. The structure of the article is clear, you mentioned the selection criteria for the 35 scientific papers presented, you presented a lot of variables that influence involvement in physical activities, the future research direction (related to the lack of studies for the interaction of the social environment and the built environment) was identified. I offer some suggestions for improving the reviewed version:

1. Manuscript citation style is correct for first sources [in square brackets, eg 2,4, 6, etc.]. Starting with source (9) - line 63 - and all the way to the end of the article......this is erroneous.

2. Figure 1/page 4 is important for identifying the criteria and procedures for selecting the analyzed articles. Maybe you can increase its size to make it clearer.

3. Table 1 has more than half the volume of information in your entire manuscript, perhaps it would be useful to transfer it to an Appendix. It is just an idea to reduce the number of pages of the study.

4. Lines 256, 259, 265, 270, 311 indicate sources not identified by the software you used to insert them into the text: (Error! Reference source not found.).

5. The page numbering is incorrect. The first 5 pages show correct numbering, but starting with table 1 these problems appear (repeat of the same page numbers). This error is also present for pages inserted at the end of table 1...........

6. Figure 2/page 35/line 452 – Insertion of figure title below it is missing.

7. Line 476 (source 89 is cited twice).

8. Are there major differences of opinion and/or common views between the studies reviewed? I'm thinking about the variations that can occur by geographic region, gender or age.

9. The results and discussion present a very large number of variables (high volume of data) that are associated with the social environment and the built environment. They have different influence on the level of involvement in physical activities (PA). Maybe you can summarize the most relevant ones.

10. The conclusions can be better correlated with the two research directions presented (lines 113-118).

Round 2

Reviewer 1 Report

The authors have clarified all my doubts and I have nothing more to say.

Thank you very much

Author Response

Thank you once again for your time, expertise, and invaluable feedback. We greatly appreciate your contributions to our manuscript, and we look forward to the opportunity of future collaboration.

Best,

Yiyu